# NEURAL VOLUMETRIC MESH GENERATOR

## ABSTRACT

Deep generative models have shown success in generating 3D shapes with different representations. In this work, we propose Neural Volumetric Mesh Generator (NVMG), which can generate novel and high-quality volumetric meshes. Unlike the previous 3D generative model for point cloud, voxel, and implicit surface, the volumetric mesh representation is a ready-to-use representation in industry with details on both the surface and interior. Generating this such highly-structured data thus brings a significant challenge. We first propose a diffusion-based generative model to tackle this problem by generating voxelized shapes with close-to-reality outlines and structures. We can simply obtain a tetrahedral mesh as a template with the voxelized shape. Further, we use a voxel-conditional neural network to predict the smooth implicit surface conditioned on the voxels, and progressively project the tetrahedral mesh to the predicted surface under regularizations. The regularization terms are carefully designed so that they can (1) get rid of the defects like flipping and high distortion; (2) force the regularity of the interior and surface structure during the deformation procedure for a high-quality final mesh. As shown in the experiments, our pipeline can generate high-quality artifact-free volumetric and surface meshes from random noise or a reference image without any post-processing. Compared with the state-of-the-art voxel-to-mesh deformation method, we show more robustness and better performance when taking generated voxels as input.

## 1 INTRODUCTION

How to automatically create high-quality new 3D contents that are accessible and editable is a key problem in visual computing. Although generative models have revealed their power on audio and image synthesis (Goodfellow et al., 2014; Kingma & Welling, 2013; Higgins et al., 2016; Goodfellow et al., 2014; Brock et al., 2018; Ho et al., 2019; Song & Ermon, 2019; Ho et al., 2020; Song et al., 2020), their performance remains limited. A major challenge of the current methods is the representation of the 3D shapes. Many generative models focus on point clouds (Fan et al., 2017; Yang et al., 2019; Cai et al., 2020; Luo & Hu, 2021a;b). However, it is non-trivial to convert point clouds to other shape representations. Another line of work (Park et al., 2019; Niemeyer & Geiger, 2021; Schwarz et al., 2020; Jain et al., 2021) directly learns to generate implicit representations, e.g., neural radiance field (NeRF) (Mildenhall et al., 2020), of shapes. However, for applications in physical simulation, the implicit representation needs to be converted into explicit representations such as meshes, which by itself is not a completely solved problem.

In this work, we consider the problem of directly generating ready-to-use volumetric meshes. Volumetric mesh is one of the most important representations of 3D shapes, which is widely adopted in computer graphics and engineering (Nieser et al., 2011; Hang, 2015; Hu et al., 2018). However, it is difficult to be generated with off-the-shelf generative models due to a number of geometric constraints (Aigerman & Lipman, 2013; Li et al., 2007; 2020; 2021; Ni et al., 2021). Without carefully handling these constraints, the generated meshes have various defects, including flipping, self-intersection, large distortion, etc. To overcome the constraints, existing methods (Wang et al., 2018; Wen et al., 2019; Gupta & Chandraker, 2020; Shi et al., 2021) for deep mesh generation usually learn deformation on a template mesh, e.g., an ellipsoid mesh, to obtain a new mesh. Unfortunately, the usage of a template mesh limits the topology (number of holes) and large deformation of the generated meshes.

Thus, we present a novel pipeline, termed Neural Volumetric Mesh Generator (NVMG), for learning generative models of volumetric meshes. Unlike focusing on designing the neural network on the mesh representation, NVMG takes a two-level hybrid approach. First, we utilize the generalization

behavior of diffusion models (Ho et al., 2020) on a voxel-based representation to obtain an initial synthesis result. We then use another neural network to predict how to perturb the initial synthesis, adding geometric details. At the second level, NVMG employs an optimization procedure to control the quality of the final mesh, addressing issues in flipped faces and distorted faces. The optimization procedure seamlessly combines with the output of neural predictions, adding the strength of neural predictions and optimization-based formulations for mesh optimization.

We empirically evaluate NVMG on unconditional volumetric mesh generation and conditional image-to-mesh generation tasks. We show that NVMG outperforms state-of-the-art point cloud generator (Zhou et al., 2021) and image-to-mesh generator (Shi et al., 2021). When we compare our mesh deformation module with the state-of-the-art voxel-to-mesh methods and Neural Dual Contouring, we obtain better results on converting generated voxel to mesh.

## 2  RELATED WORK

**3D Shape Generation**  Generating 3D shapes is a key challenge in computer vision and computer graphics (Hartley & Zisserman, 2003; Moons et al., 2010). Traditional methods focus on reconstructing 3D shapes from multiple views (Hartley & Zisserman, 2003; Furukawa & Ponce, 2009) using geometric cues. With the power of deep learning, researchers have advanced the field significantly with data-driven methods. Through the prior knowledge captured by the deep neural networks, current algorithms can generate point clouds from a single image (Fan et al., 2017; Yang et al., 2019; Achlioptas et al., 2018; Luo & Hu, 2021b;a; Zhou et al., 2021), improve multi-view reconstruction (Yao et al., 2018; Chen et al., 2019), and create high-quality implicit sur-

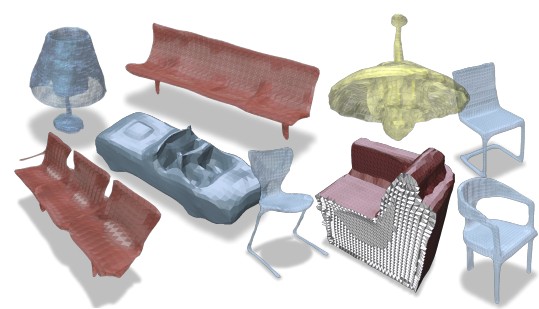

Figure 1: A gallery of generated volumetric meshes. Both the surface and the interior of volumetric meshes generated by NVMG are artifact-free.

faces (Mildenhall et al., 2020; Niemeyer & Geiger, 2021; Genova et al., 2020). Different from this work, our work focuses on an even harder 3D representation, the volumetric mesh. Instead of generating the point position or signed distribution in the 3D space, we need to provide the vertices position and the connection structure for face relations for both the surface and interior. The complex relationship makes the generative model extremely hard to learn.

**Mesh Generation with Deep Neural Networks**  Despite the generality of deep learning, how to incorporate deep neural networks efficiently in mesh generation remains an open problem. The main difficulty is that a valid mesh must satisfy a series of physical constraints. To deal with these constraints, the majority of previous works use a mesh template and train a neural network to deform the template to obtain the mesh of interest (Wang et al., 2018; Wen et al., 2019; Gupta & Chandraker, 2020; Shi et al., 2021; Kanazawa et al., 2018; Pan et al., 2019; Uy et al., 2020). However, the generated meshes are limited by the pre-defined template in several aspects, e.g., topology and the magnitude of deformation. Another branch of work is to learn implicit field to reconstruct the mesh from the point clouds or voxel prior (Venkatesh et al., 2021; Chen et al., 2022; Shen et al., 2021b; Chen & Zhang, 2019; 2021; Gao et al., 2020; Remelli et al., 2020; Chen et al., 2020; 2021; Chibane et al., 2020b; Sitzmann et al., 2020; Williams et al., 2019). These works rely on the Marching Cube or variant of the marching cube algorithm to convert the learned implicit representation to the surface. However, these methods mainly have three issues. First, most methods need to know the information of the point cloud on the surface for the reconstruction, which indicates that they are not suitable for generating novel shapes. Second, converting implicit surface to mesh is not robust on the shape topology and may cause part and hole losses when converting to mesh. Third, the implicit surface representation can not model the shape interior. PolyGen (Nash et al., 2020) uses sequential model to progressively generate vertex and faces one by one, however, this kind of auto regressive model using transformer structures is costly and exhibits only limited generalization behavior. GET3D (Gao et al., 2022) generates high-quality 3D textured meshes bridging recent advances in the differentiable surface modeling, differentiable rendering as well as 2D Generative Adversarial Networks.

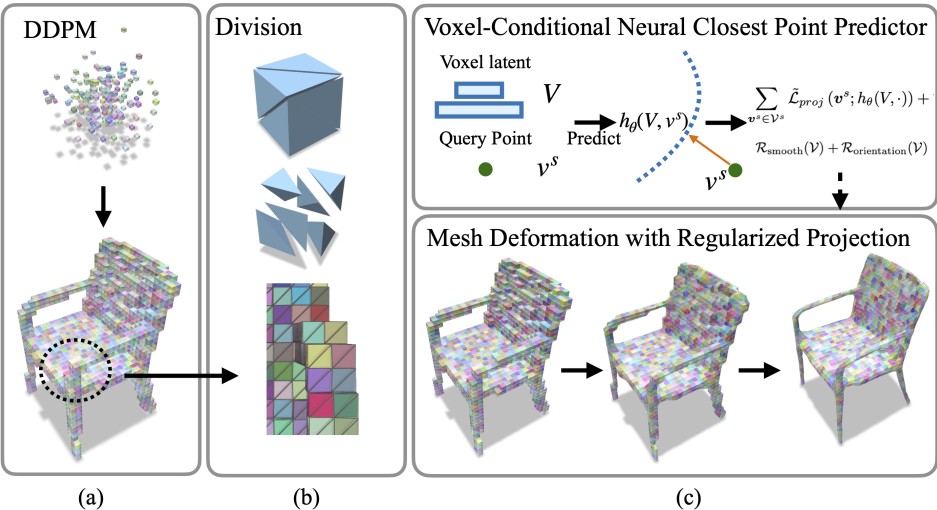

Figure 2: NVMG combines three modules. (a) A generative model for generating voxelized shapes from noise distribution; (b) Voxel division for the initial volumetric mesh; (c) Physically correct mesh deformation with voxel-conditional neural closest point predictor and regularized projection.

**Hybrid representations for shape synthesis.** Choosing a suitable representation for shape synthesis is fundamentally challenging. Several recent works have studied combing hybrid 3D representations for geometry synthesis. Zhang et al. (2020) combine a point-based representation and an image-based representation for scene synthesis. Yang et al. (2021) combine neural networks and parametric priors of object relations for scene synthesis. Shen et al. (2021a) combine implicit representation and an explicit tetrahedral representation for synthesizing shape details from a coarse volumetric grid. In contrast, our approach combine both hybrid representations and procedures for mesh synthesis which prioritize high-quality shape details and mesh quality.

## 3 NEURAL VOLUMETRIC MESH GENERATOR

This section presents the technical details of our neural volumetric mesh generator (or NVMG). As illustrated in Figure 2, NVMG combines three newly proposed modules: voxel generation with diffusion models, voxel-conditional neural closest point predictor, and physically robust mesh deformation with a regularized projection. These modules nicely combine the strength of different approaches. Specifically, diffusion-based methods show great generalization behavior on the global shape, and it provides an initial tetrahedral mesh. The neural closet point predictor adds surface details. while the mesh deformation module promotes the mesh quality.

### 3.1 VOXEL GENERATION WITH DIFFUSION MODELS

The first module of NVMG is voxel generation. We exploit denoising diffusion probabilistic model (DDPM) (Ho et al., 2020) for this task due to its simplicity and generalization behavior.

**DDPM** The goal of DDPM is to learn a parameterized Markov chain that can generate novel samples after finite time. In the forward direction $q(\boldsymbol{x}_{0:T})$ of the chain, a real data sample $\boldsymbol{x}_0$ is gradually diffused to a sample $\boldsymbol{x}_T$ from the standard Gaussian noise by adding noise in each time step. DDPM then trains a conditional denoising model $p_\phi(\boldsymbol{x}_{t-1}|\boldsymbol{x}_t)$ that runs in the reverse direction, which reduces the noise on $\boldsymbol{x}_t$ in each time step $t$ to obtain a synthetic sample.

Formally, we have two Markov chains in opposite directions, the *forward process* $q(\boldsymbol{x}_{0:T})$ and the *reverse process* $p_\phi(\boldsymbol{x}_{0:T})$,

$$q(\boldsymbol{x}_{0:T}) = q(\boldsymbol{x}_0)\Pi_{t=1}^T q(\boldsymbol{x}_t|\boldsymbol{x}_{t-1}), \quad p_\phi(\boldsymbol{x}_{0:T}) = p(\boldsymbol{x}_T)\Pi_{t=1}^T p_\phi(\boldsymbol{x}_{t-1}|\boldsymbol{x}_t). \quad (1)$$

Here, the transition probabilities are defined as Gaussian distributions,

$$q(\boldsymbol{x}_t|\boldsymbol{x}_{t-1}) = \mathcal{N}(\boldsymbol{x}_t; \sqrt{1-\beta_t}\boldsymbol{x}_{t-1}, \beta_t\boldsymbol{I}), \quad p_\phi(\boldsymbol{x}_{t-1}|\boldsymbol{x}_t) = \mathcal{N}(\boldsymbol{x}_{t-1}; \boldsymbol{\mu}_\phi(\boldsymbol{x}_t, t), \beta_t\boldsymbol{I}). \quad (2)$$

In practice, DDPM choose the following parameterization, $\boldsymbol{\mu}_\phi(\boldsymbol{x}_t, t) = \frac{1}{\sqrt{\alpha_t}}\left(\boldsymbol{x}_t - \frac{\beta_t}{\sqrt{1-\bar{\alpha}_t}}\boldsymbol{\epsilon}_\phi(\boldsymbol{x}_t, t)\right)$, where $\alpha_t = 1 - \beta_t$ and $\bar{\alpha}_t = \Pi_{i=1}^t \alpha_i$. $\beta_t$, which controls

the magnitude of noise, gradually decreases to $0$ as $t$ approaches $0$. The training objective is to maximize the evidence lower bound (ELBO). The derivation leads to a simple loss function,

$$\mathcal{L}_{ddpm} = \mathbb{E}_{\boldsymbol{x}_0, t, \boldsymbol{\epsilon}} \left[ ||\boldsymbol{\epsilon} - \boldsymbol{\epsilon}_\phi(\sqrt{\bar{\alpha}_t}\boldsymbol{x}_0 + \sqrt{1 - \bar{\alpha}_t}\boldsymbol{\epsilon}, t)||^2 \right], \tag{3}$$

where $\boldsymbol{x}_0 \sim \mathcal{D}_{train}$ is a data point sampled from the training set, $t$ is uniformly sampled from $1$ to $T$, and $\boldsymbol{\epsilon} \sim \mathcal{N}(\mathbf{0}, \boldsymbol{I})$. The neural network $\boldsymbol{\epsilon}_\phi(\boldsymbol{x}_t, t)$ learns to estimate the noise from current observation $\boldsymbol{x}_t$, then subtract it to obtain the new estimation of mean in time step $t - 1$. After the training process, one can generate new samples through simulating the *reverse process*. Starting from $\hat{\boldsymbol{x}}_T \sim \mathcal{N}(\mathbf{0}, \boldsymbol{I})$, the reverse process recursively draws $\hat{\boldsymbol{x}}_{t-1}$ from $p_\phi(\hat{\boldsymbol{x}}_{t-1}|\hat{\boldsymbol{x}}_t)$ until a generated sample $\hat{\boldsymbol{x}}_0$ is obtained. Readers can refer to (Ho et al., 2020) for more details of DDPM.

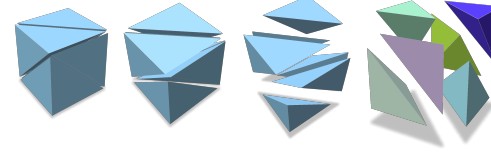

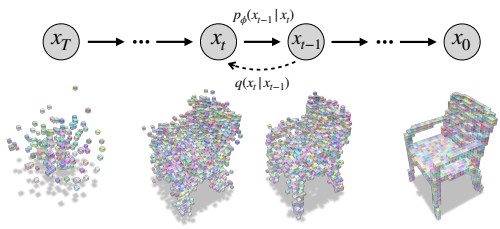

Figure 4: A visualization of transforming voxel to tetrahedral mesh. Each voxel can be split into 6 tetrahedra.

Figure 3: Generating voxelized shapes with DDPM.

**Voxel Generation with DDPM** In our scenario, shapes are normalized to a unit cube with $r \times r \times r$ voxels. A voxel representation of a shape is encoded by a binary tensor $V \in \{0, 1\}^{r \times r \times r}$, where $V(x, y, z) = 0/1$ means that the $(x, y, z)$-th voxel is unfilled/filled. The training set for voxel generation is created by discretizing the original meshes in the training set to their voxel representations with specific resolution. We adopt a similar 3D convolutional neural network as in (Zhou et al., 2021) to parameterize $\boldsymbol{\epsilon}_\phi$, with several small modifications to accommodate it to the voxel generation task. Details can be found in the supp. material.

**Tetrahedral Mesh from Voxel Division** After the coarse voxel representation of the shape is generated, we then divide it into tetrahedral mesh for later modules of NVMG. A tetrahedron is a pyramid-like polyhedron, served as a basic volumetric element, with four vertices and four triangular faces. A hexahedron is a polyhedron, with eight vertices and six quadrilateral faces. In our method, we split each hexahedron into 6 tetrahedra as in Figure 4 to achieve the face-to-face consistency across the tetrahedral mesh.

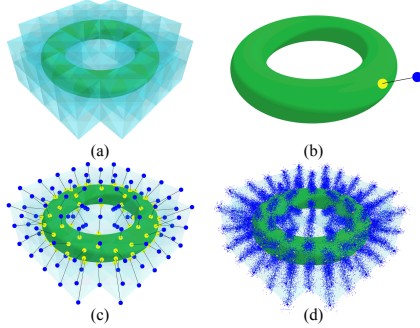

Specifically, we first represent the voxel-based representation as a hexahedral mesh $\mathcal{H} = (\mathcal{V}, \mathcal{F}_{quad}) = (\underset{c}{\cup}\{\boldsymbol{v}_c^i\}_{i=1}^{i=8}, \underset{c}{\cup}\{\boldsymbol{f}_c^j\}_{j=1}^{j=6})$, where $\mathcal{V}$ is the vertex set, $\mathcal{F}_{quad}$ is the quadrilateral face set, $\boldsymbol{v}_c^i$ is the $i$-th vertex of the $c$-th cube, and $\boldsymbol{f}_c^j$ is the $j$-th quadrilateral face of the $c$-th cube. By splitting each cube mesh into 6 tetrahedra, we can further extract a tetrahedral mesh $\mathcal{T} = (\mathcal{V}, \mathcal{F}_{tri}) = (\underset{c}{\cup}\overset{6}{\underset{k=1}{\cup}}\{\boldsymbol{v}_{c,k}^i\}_{i=1}^{i=4}, \underset{c}{\cup}\overset{6}{\underset{k=1}{\cup}}\{\boldsymbol{f}_{c,k}^j\}_{j=1}^{j=4})$, where $\mathcal{V}$ is the same vertex set as the hexahedral mesh $\mathcal{H}$, $\mathcal{F}_{tri}$ is the triangular face set, $\boldsymbol{v}_{c,k}^i$ is the $i$-th vertex of the $k$-th tetrahedron in

Figure 5: (a) Voxelization of torus. (b) Blue query point $\boldsymbol{x}$ and its yellow closest point $\text{CP}(\boldsymbol{x})$. (c) Query from $\mathcal{V}^s$ to torus. (d) Training samples.

the $c$-th cube, and $\boldsymbol{f}_{c,k}^j$ is the $j$-th triangular face of the $k$-th tetrahedron in the $c$-th cube. For volumetric meshes, as there are interior meshes, we use $\mathcal{V}^s$ to denote the set of the boundary vertices on the surface.

## 3.2 A VOXEL-CONDITIONAL NEURAL CLOSEST POINT PREDICTOR

The voxelized shape and its corresponding tetrahedral mesh is a coarse representation the shape of interest. The second module of NVMG employs a neural network synthesize surface details from the voxelized shape. This is done by fitting a neural network $h_\theta(V, \boldsymbol{x}) : \{0, 1\}^{r^3} \times \mathbb{R}^3 \to \mathbb{R}^3$ to mimic the closest point function of each training surface $S$ which maps any point $\boldsymbol{x} \in \mathbb{R}^3$ to its closest point

on $S$:

$$\mathrm{CP}(\boldsymbol{x}) = \arg\min_{\boldsymbol{p}\in S} \|\boldsymbol{p}-\boldsymbol{x}\|_2. \tag{4}$$

Note that during generation, the surface $S$ is implicitly defined by $S = \{\boldsymbol{x}\in\mathbb{R}^3 \mid h_\theta(V,\boldsymbol{x})-\boldsymbol{x}=\mathbf{0}\}$. The closest point function becomes discontinuous at points on the medial axis. Therefore, the closest point prediction network only applies for points that are close to the final surface. This is another motivation of using the voxelized shape as the initial mesh reconstruction.

**Data preparation**    To train the prediction network $h_\theta(V,\boldsymbol{x})$, we first normalize each mesh in the training dataset to a unit bounding box, voxelize it and extract the mesh $\mathcal{T}$ as described before. Then, we track the boundary $\partial\mathcal{T}$, which is a union of triangular faces $\mathcal{F}_{tri}^s$ that forms a watertight surface mesh $\mathcal{T}^s = (\mathcal{V}^s, \mathcal{F}_{tri}^s)$. Next, we calculate the closest point $\boldsymbol{p}^s$ on the ground truth object surface to each vertex $\boldsymbol{v}^s \in \mathcal{V}^s$. Instead of sampling spatial points near the surface aggressively as deep implicit function(Park et al., 2019), we sample points around line segments $\{(1-\alpha)\boldsymbol{p}^s + \alpha\boldsymbol{v}^s | \alpha \in [0,1]\}$ as depicted in Figure 5. Therefore, our sampling area for the closest function is tied with the output of the first module voxel shape, i.e. a band within the voxel cube that wraps up the target surface.

**Discussion**    A recent work, CSPNet (Venkatesh et al., 2021) also predicts the closest point for surface reconstruction. However, CSPNet need to take the ground truth point cloud and its normal as input to calculate the closest point, which are not available for the generation task.

### 3.3    Physically Robust Mesh Deformation with Regularized Projection

We proceed to introduce the third module of NVMG, which takes the output of the first two modules as input and output a deformed mesh. A straightforward approach is to directly project each vertex of the volumetric mesh to its closest point inferred by the second module $h_\theta$:

$$\boldsymbol{v}^s \leftarrow h_\theta(V,\boldsymbol{v}^s), \quad \forall \boldsymbol{v}^s \in \mathcal{V}^s. \tag{5}$$

However, this naive projection does work well in practice due to three issues. First, the prediction from the neural network can be imprecise, creating bumps on the resulting mesh. Second, the projections can be highly non-uniform on the surface, causing highly distorted mesh. Third, the projections are independent with each other, leading to artifacts like flipping.

Therefore, we introduce an optimization formulation to jointly optimize the locations of projected vertices. The objective terms are designed to promote several generic conditions about a high-quality volumetric mesh:

$$\mathcal{L}(\mathcal{V}) = \sum_{\boldsymbol{v}^s\in\mathcal{V}^s} \mathcal{L}_{proj}\left(\boldsymbol{v}^s; h_\theta(V,\cdot)\right) + \mathcal{R}(\mathcal{V}), \tag{6}$$

where $\mathcal{L}_{proj}\left(\boldsymbol{v}^s; h_\theta(V,\cdot)\right) = \|\boldsymbol{v}^s - h_\theta(V,\boldsymbol{v}^s)\|_2$ is the data term and $\mathcal{R}(\mathcal{V})$ is the regulatization term. The data term prioritizes that the overall shape of the projection stay close to the output of the second module. The regularization term $\mathcal{R}(\mathcal{V})$ promotes the mesh quality, which is designed to be differentiable so that it can be easily optimized with any gradient-based optimizer. In the following, we introduce the regularization term $\mathcal{R}$ and the data term $\mathcal{L}_{proj}$ in details.

**1) Smooth term for uniform structure**    The first regularization term $\mathcal{R}_{\mathrm{smooth}}$ promotes the smoothness of the resulting tetrahedral mesh $\mathcal{T}$:

$$\mathcal{R}_{\mathrm{smooth}}(\mathcal{V}) = \lambda_a \mathcal{R}_{\mathrm{edge}}(\mathcal{V}, \mathcal{F}_{tri}) + \lambda_b \mathcal{R}_{\mathrm{laplacian}}(\mathcal{V}, \mathcal{F}_{tri}) + \lambda_c \mathcal{R}_{\mathrm{normal}}(\mathcal{V}^s, \mathcal{F}_{tri}^s). \tag{7}$$

Here $\mathcal{R}_{\mathrm{edge}}$ penalizes extremely long edges; $\mathcal{R}_{\mathrm{laplacian}}$ is the graph-based Laplacian term; $\mathcal{R}_{\mathrm{normal}}$ enforces normal consistency among adjacent faces on the surface. Note that, $\mathcal{R}_{\mathrm{edge}}$ and $\mathcal{R}_{\mathrm{laplacian}}$ are defined over the whole tetrahedral mesh, which differs from (Wang et al., 2018). $\lambda_a, \lambda_b$ and $\lambda_c$ are hyper-parameters, and we determine them via cross-validation.

**2) Orientation term to prevent defects**    Another important property of a tetrahedral is to avoid vanishing face volumes and flips. Flips break local injectivity(Abulnaga, 2018) and cause salient artifacts. Therefore, we introduce an orientation term which prevent each tetrahedral face from approaching zero volume and thus enforces local injectivity:

$$\mathcal{R}_{\mathrm{orientation}}(\mathcal{V}) = \sum_{c,k} l(\boldsymbol{M}^{c,k}), \tag{8}$$

$$l(\boldsymbol{M}^{c,k}) = \begin{cases} -(\det(\boldsymbol{M}^{c,k})-v_0)^2 \log(\frac{\det(\boldsymbol{M}^{c,k})}{v_0}) & , \det(\boldsymbol{M}^{c,k}) \le v_0 \\ 0 & , \det(\boldsymbol{M}^{c,k}) > v_0 \end{cases}, \tag{9}$$

where $\boldsymbol{M}^{c,k} \in \mathbb{R}^{4 \times 4}$ is the matrix stacked by the homogeneous coordinates of four vertices of the $k$-th tetrahedron in the c-th cube. The vertices order in $\boldsymbol{M}^{c,k}$ satisfy the right hand rule convention such that every tetrahedron in the initial mesh has a positive volume. Note that $l(\cdot)$ is a robust $C^2$-function, which is widely used in physical simulation (Li et al., 2020; 2021; Ni et al., 2021). $v_0$ is a constant thresholding set to activate the penalty, ensuring a minimum volume of tetrahedra. By this orientation-preserving term, each tetrahedron keeps a positive volume through the optimization procedure.

**3) Robust Projection for distortion suppression**    We empirically observed that, merely pulling the mesh vertices to the neural prediction of the closest point may result in colliding vertices and mesh faces with large distortion. To address this issue, we modify $\mathcal{L}_{proj}$ using a robust version:

$$\tilde{\mathcal{L}}_{proj}\left(\boldsymbol{v}^s; h_\theta(V, \cdot)\right) = \|\boldsymbol{v}^s - h_\theta(V, \boldsymbol{v}^s) + k\boldsymbol{n}\|_2, \tag{10}$$

where $\boldsymbol{n} \sim \mathcal{N}(0, 1)$ is a random Gaussian noise, $k$ is a coefficient that gradually decays to 0 as optimization proceeds. During the optimization procedure, the 'lucky' vertices arrive at the surface earlier, while the 'unlucky' vertices arrive later. Thus the collisions are avoided, and distortions of the resulting mesh are significantly reduced.

**Putting them together**    By combining $\tilde{\mathcal{L}}_{proj}$, $\mathcal{R}_{\text{smooth}}$ and $\mathcal{R}_{\text{orientation}}$, our final optimization problem is,

$$\mathcal{L}(\mathcal{V}) = \sum_{\boldsymbol{v}^s \in \mathcal{V}^s} \tilde{\mathcal{L}}_{proj}\left(\boldsymbol{v}^s; h_\theta(V, \cdot)\right) + \mathcal{R}_{\text{smooth}}(\mathcal{V}) + \mathcal{R}_{\text{orientation}}(\mathcal{V}). \tag{11}$$

After obtaining the voxelized shape and the tetrahedral mesh from Sec. 3.1, we optimize Eq. equation 11 to obtain the final volumetric mesh.

# 4    EXPERIMENTAL EVALUATION

We begin with the unconditional mesh generation setting in Section 4.1. We then evaluate NVMG for mesh generation from image inputs in Section 4.2. Section 4.3 and Section 4.4 present an analysis of the mesh quality of NVMG and its application in shape editing, respectively.

**Implementation details** The voxel generation module uses a 8-layer network on a $32^3$ grid. The denoising procedure uses a linear noise schedule from $0.02$ to $0.0001$, where number of steps is $1000$. We train this module 500 epochs with the learning rate $2e^{-3}$. The neural closest point module employs a multi-scale voxel network inspired by IF-Net (Chibane et al., 2020a). This network has 6-layer 3D convolutions that extract the features from layers at resolutions $32^3$, $16^3$, and $8^3$. The extracted features are fed into a fully connected layer to predict the closest points. The mesh deformation module uses 60 iterations. We use $4 \times$ RTX2080Ti GPUs for training the neural modules. Please refer to the supp. material for more details.

## 4.1    UNCONDITIONAL MESH GENERATION

We first evaluate unconditional mesh generation which takes a latent code from a random distribution. Baseline approaches include state-of-the-art point cloud generation methods including PointFlow (Yang et al., 2019), ShapeGF (Cai et al., 2020), Point cloud diffusion model (Luo & Hu, 2021a), and PVD (Zhou et al., 2021), and Neural Dual Contouring (or NDC) (Chen et al., 2022), a state-of-the-art voxel-to mesh method.

**Dataset**    We train this pipeline using ShapeNet Chair and Airplane categories, which are the two most representative subsets among baseline approaches (Achlioptas et al., 2018; Yang et al., 2019; Cai et al., 2020; Luo & Hu, 2021a; Zhou et al., 2021; Chen et al., 2022).

**Evaluate Metric**    The same as (Zhou et al., 2021), we evaluate shape generation quality using 1-NN under both Chamfer distance and Earth Mover's Distance. Supp. material reports more evaluations. Supp. material reports additional evaluations under Minimum Matching Distance (MMD) and Coverage (COV), which are two other commonly used metrics.

**Result**    We see from Table 1, NVMG gets a comparable and even better performance compared with the state-of-the-art baseline approaches. Note that this is encouraging as NVMG does not optimize the point distribution directly. Figure 7 further shows that one of the advantages of NVMG over prior works in the sense that the generated shape is smooth without outliers.

| Base rep. | Method | Airplane | | Chair | |
|---|---|---|---|---|---|
| | | CD | EMD | CD | EMD |
| Voxel | Voxel-VAE (Brock et al., 2016) | 94.31 | 95.43 | 91.21 | 91.56 |
| | Voxel-GAN (Wu et al., 2016) | 86.22 | 79.59 | 74.42 | 81.29 |
| | Vox-diffusion (Zhou et al., 2021) | 99.75 | 98.13 | 97.12 | 96.74 |
| | NVMG (Voxel generator) | **82.14** | **71.01** | **68.92** | **68.84** |
| Point Cloud | l-GAN (Achlioptas et al., 2018) | 87.30 | 93.95 | 68.58 | 83.84 |
| | PointFlow (Yang et al., 2019) | 75.68 | 70.74 | 62.84 | 60.57 |
| | DPF-Net (Klokov et al., 2020) | 75.18 | 65.55 | 62.00 | 58.53 |
| | SoftFlow (Kim et al., 2020) | 76.05 | 65.80 | 59.21 | 60.05 |
| | ShapeGF (Cai et al., 2020) | 80.00 | 76.17 | 68.96 | 65.48 |
| | Diffusion (Luo & Hu, 2021a) | 74.14 | 65.12 | 56.24 | 54.28 |
| | PVD (Zhou et al., 2021) | 73.82 | 64.81 | 56.26 | 53.32 |
| Mesh | PolyGen (Nash et al., 2020) | 81.51 | 72.32 | 69.12 | 63.90 |
| | NVMG (w/ NDC (Chen et al., 2022)) | 76.41 | 67.20 | 58.75 | 55.38 |
| | NVMG | **73.41** | **64.29** | **56.12** | **53.30** |

Table 1: Result compared with various point cloud generation baselines. CD: Chamfer distance. EMD: Earth Mover's Distance. Lower score means better generation quality and diversity.

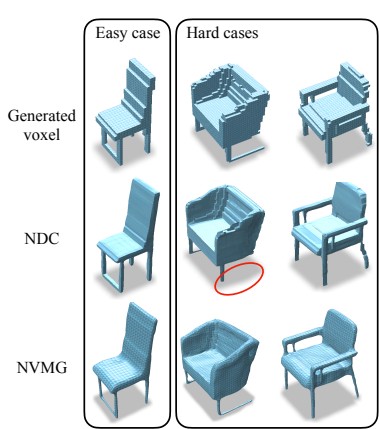

Figure 6: Mesh deformation uses our method and NDC. We notice the heavy artificial and even a missing leg (column 2 in red circle) on the NDC mesh for the hard cases.

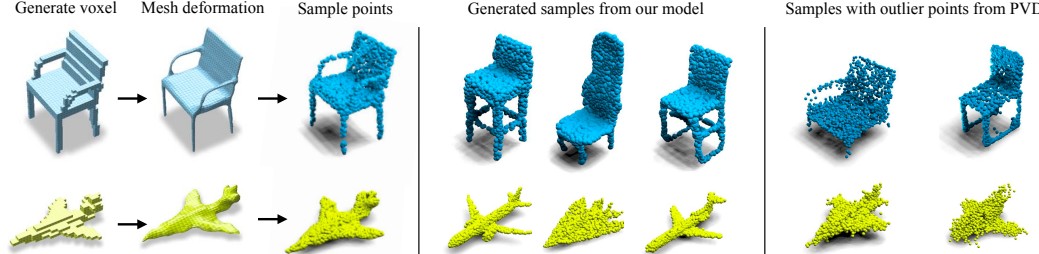

Figure 7: Comparison between NVMG and PVD (Zhou et al., 2021), a state-of-the-art point-based generator. For fair comparisons, we convert the output mesh of NVMG into a point cloud.

#### 4.1.1 ABLATION STUDY

To show our voxel generator and voxel to mesh deformation are carefully designed for this pipeline and not easily to be replaced, we compare each parts with the existing literature and introduce the ablation study results in Table 1.

**Effectiveness of the voxel generator** First, we validate our voxel diffusion model is powerful enough to generate good-enough intermediate voxel representation. As we show in the first voxel representation section in the Table 1, we compare several voxel generator and calculate the metric of the generation quality. We reimplement Voxel-VAE (Brock et al., 2016) and Voxel-GAN (Wu et al., 2016) due to the official code base is too old and use the Vox-Diffusion result from PVD (Zhou et al., 2021). We see that NVMG voxel generator part greatly exceed other voxel generators and thus provide a good-quality of intermediate voxels for the following deformation parts.

**Robustness of the mesh deformeration** There are some brilliant voxel to mesh designs in recent work including NDC (Chen et al., 2022) and DMtet (Shen et al., 2021b). Because DMtet requires additional surface points information for the voxel to mesh reconstruction, which is not suitable for our generated voxels, we choose NDC and train it in the same training set for a relative fair comparison. We compare our mesh deformation with NDC by taking our generated voxel samples as input. From Table 1 last two lines, we observed a higher performance of our method compared with NDC, indicating our method are more robust in converting the generated voxel samples into mesh surface. In Figure 6, we visualize the generated mesh samples. In easy case, NDC performs similarly to our method. However, generated voxels may contain irregular noise. so in the hard cases, NDC may shows more artificial on the surface. In addition, we also observed a missing leg on the second column from the NDC's result, showing the insatiability of the marching cube variant algorithm with noisy voxel samples as input. The above result shows our design of the deformation-based method and robustness regularization loss works better in our proposed pipeline comparing with the off-the-shell marching cube based voxel to mesh methods.

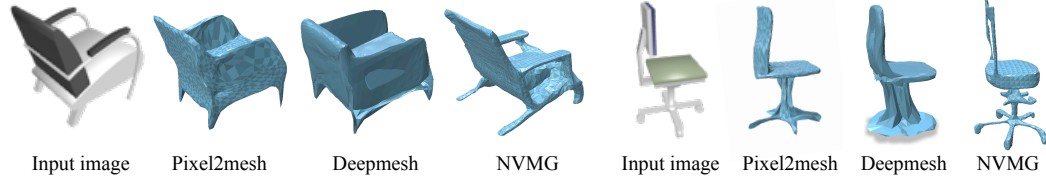

Input image    Pixel2mesh    Deepmesh    NVMG    Input image  Pixel2mesh  Deepmesh  NVMG

Figure 8: Qualitative comparisons between NVMG and two baseline approaches Pixel2mesh (Wang et al., 2018) and Deepmesh (Pan et al., 2019) for the task of single-view shape reconstruction.

## 4.2 IMAGE CONDITIONAL SHAPE GENERATION.

We proceed to evaluate NVMG for the task of image to mesh generation. To this end, we use ResNet-18 (He et al., 2016) to extract image features and fit the extracted features into NVMG for reconstruction. We compare this approach with state-of-the-art image to mesh based approaches, including Pixel2mesh (Wang et al., 2018), GEOMetric (Smith et al., 2019), and DeepMesh (Pan et al., 2019).

**Dataset and Metric.** We use the ShapeNetRender dataset (Choy et al., 2016). We select three categories, i.e., chair, bench and table, which are the most common yet challenging categories among prior works. Same as the previous works, we uniformly sample 10,000 points on the reconstructed mesh and compare them with the ground-truth surface using the Chamfer distance.

**Result.** Table 2 that NVMG outperforms state-of-the-art approaches consistently. Figure 8 presents qualitative results of NVMG and two top performing baselines Pixel2Mesh and Deepmesh. We can see that NVMG offers reconstructions that preserve more shape details and thin geometric features than baseline approches.

|       | P2M   | GEOMetrics | DeepMesh | NVMG  |
|-------|-------|------------|----------|-------|
| chair | 0.610 | 0.823      | 0.514    | 0.471 |
| table | 0.498 | 0.797      | 0.404    | 0.342 |
| bench | 0.624 | 0.690      | 0.516    | 0.439 |

Table 2: Comparisons between NVMG and baseline approaches under the Chamfer distance.

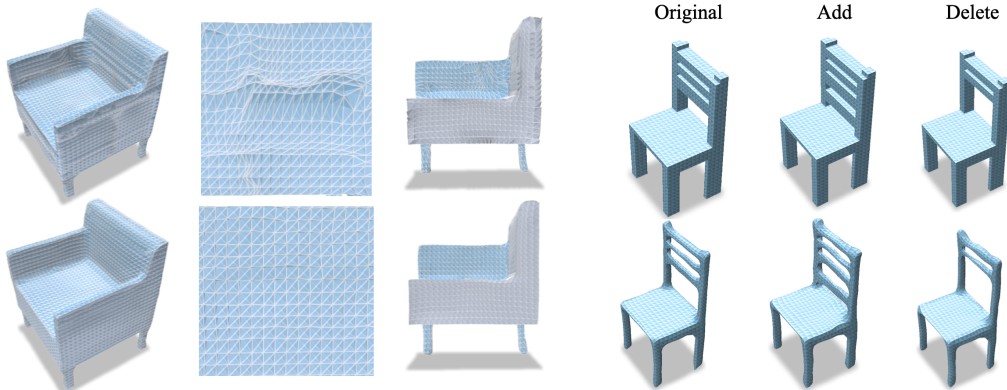

Original       Add       Delete

Figure 9: Robust projection regularization improves the quality of the reconstructed mesh. (Top) Without robust projection (Bottom) With robust projection.

Figure 10: An application in shape editing where the user edits the coarse voxel shape and the final shape is updated accordingly

## 4.3 ABLATION STUDY ON THE LOSS TERMS

A desired property of surface or volumetric mesh reconstruction shall be free of flips and distorted cells. Figure 11 shows the effects of different combinations of regularization terms. We detect flips by calculating signs of their oriented volumes and color them in red. The quality of a cell is assessed by the aspect ratio (Parthasarathy et al., 1994), which is defined as as the ratio between its circumradius

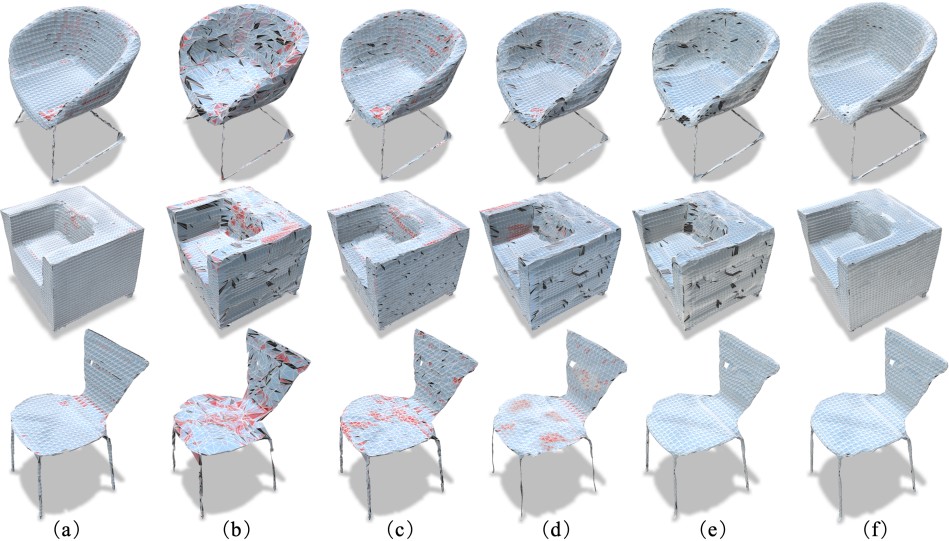

Figure 11: We show the ablation study for our method. Red color marks flipped tets; black color marks distorted triangle with an aspect ratio larger than 2.6. (a) One-step mapping by neural closest point function. (b) Only projection term. (c) Robust projection term. (d) Projection and smooth terms. (e) projection, smooth and orientation terms. (f) Robust projection, smooth and orientation terms.

and two times its inradius. We color the surface triangles whose aspect ratios are bigger than 2.6 in black.

As shown in Figure 11(a)(b), only using the surface term yields good approximations of the underlying surface but can have flipped tests and highly distorted faces. Figure 11(e)(f) shows that enforcing the orientation term significantly reduce the number of flipped tets. Moreover, Figure 11(c)(f) shows that the robust projection term dramatically reduces the number of distorted triangle faces. Furthermore, as the orientation term also penalizes volume lower than an allowed minimal threshold, it prevents the thin structure from vanishing and thus promotes physical feasibility. This can be seen by comparing the skinny chair legs in Figure 11(e)(f) and those in Figure 11(d). Furthermore, many self-intersecting triangles and tets occur without the regularization terms (See Figure 11(a)). Please refer to the supp. material for more ablation study results.

## 4.4 SHAPE EDITING

Voxel, as an easy-to-edit 3D representation, has already been used in 3D creation in art and games. We can manually apply editing on the generated mesh to create a user-preferred shape easily. As we show in Figure 10, we can remove or add details on the chair back under the voxel-based representation, and our deformation algorithm can provide the desired mesh based on the generative result with editing. The final mesh automatically synthesizes geometric details. Due to the space limitation, we show more editing cases including shape assemble and mixture in Appendix.

## 5 CONCLUSION AND LIMITATIONS

In this paper, we propose Neural Volumetric Mesh Generator (NVMG), a novel baseline for learning a generative model for generating volumetric mesh. Empirically, we show our pipeline can generate volumetric meshes with high-fidelity and physical robustness. One limitation of our approach is on the limited resolution of the voxel-based representation. We plan to address this issue by using Octrees in the future. Another limitation is that the three modules are not trained end-to-end. As a future direction, we plan to address this issue by developing losses on the local minimums of the deformation energy, which enable end-to-end learning.

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

## A   MORE DETAILS ON METHODOLOGY

### A.1   OJECTIVE FUNCTION

**Smooth term**

$$\mathcal{R}_{\text{smooth}}(\mathcal{V}) = \lambda_a \mathcal{R}_{\text{edge}}(\mathcal{V}, \mathcal{F}_{tri}) + \lambda_b \mathcal{R}_{\text{laplacian}}(\mathcal{V}, \mathcal{F}_{tri}) + \lambda_c \mathcal{R}_{\text{normal}}(\mathcal{V}^s, \mathcal{F}_{tri}^s). \tag{12}$$

Let $\mathcal{E}$ be the set of unordered edges in the volumetric mesh.

$$\mathcal{R}_{edge}(\mathcal{V}, \mathcal{F}_{tri}) := \sum_{(i,j) \sim \mathcal{E}} \| \boldsymbol{v}_i - \boldsymbol{v}_j \|_2^2$$

Recall our definition that $\mathcal{V}$ and $\mathcal{F}$ are vertices and faces set of the volumetric mesh, while $\mathcal{V}^s$ and $\mathcal{F}^s$ are vertices and faces set on the surface mesh. $\mathcal{F}$ and $\mathcal{F}^s$ can be viewed as the graph that records the connectivity of volumetric mesh(surface mesh included) and surface mesh(surface mesh only) respectively. Hence, the volume neighborhood and the surface neighborhood of $i$-th vertex can be drawn from graph $\mathcal{F}$ and graph $\mathcal{F}^s$, denoted as $\mathcal{N}_i$ and $\mathcal{N}_i^s$ respectively. Then we define $\boldsymbol{\delta}_i = \frac{1}{|\mathcal{N}_i|} \sum_{j \sim \mathcal{N}_i} \boldsymbol{v}_j, \boldsymbol{\delta}_i^s = \frac{1}{|\mathcal{N}_i^s|} \sum_{j \sim \mathcal{N}_i^s} \boldsymbol{v}_j^s$, and give

$$\mathcal{R}_{laplacian}(\mathcal{V}, \mathcal{F}_{tri}) := \alpha \sum_{\boldsymbol{v}_i \in \mathcal{V}} \| \boldsymbol{\delta}_i - \boldsymbol{v}_i \|_2^2 + \beta \sum_{\boldsymbol{v}_i^s \in \mathcal{V}^s} \| \boldsymbol{\delta}_i^s - \boldsymbol{v}_i^s \|_2^2,$$

where $\alpha = 1$ and $\beta = 0.5$.

The normal consistency loss is constructed under the assumption that only two faces share an edge, which is always satisfied by our meshes. Let $\boldsymbol{n}_i$ denotes the normal of $i$-th face. We have

$$\mathcal{R}_{normal}(\mathcal{V}^s, \mathcal{F}_{tri}^s) = \sum_{f_i^s \sim f_j^s} 1 - cos(\boldsymbol{n}_i, \boldsymbol{n}_j).$$

$f_i^s \sim f_j^s$ represents the $i$-th face and the $j$-th face on the surface sharing the same edge.

**Orientation term**

$$\mathcal{R}_{\text{orientation}}(\mathcal{V}) = \sum_{c,k} l(\boldsymbol{M}^{c,k}), \tag{13}$$

$$l(\boldsymbol{M}^{c,k}) = \begin{cases} -(\det(\boldsymbol{M}^{c,k}) - v_0)^2 \log(\frac{\det(\boldsymbol{M}^{c,k})}{v_0}) & , \det(\boldsymbol{M}^{c,k}) \leq v_0 \\ 0 & , \det(\boldsymbol{M}^{c,k}) > v_0 \end{cases}, \tag{14}$$

$$\det(\boldsymbol{M}^{c,k}) = \begin{vmatrix} x_0^{c,k} & y_0^{c,k} & z_0^{c,k} & 1 \\ x_1^{c,k} & y_1^{c,k} & z_1^{c,k} & 1 \\ x_2^{c,k} & y_2^{c,k} & z_2^{c,k} & 1 \\ x_3^{c,k} & y_3^{c,k} & z_3^{c,k} & 1 \end{vmatrix} \tag{15}$$

$(x_i^{c,k}, y_i^{c,k}, z_i^{c,k})$ is the coordinate of the $i$-th vertex of the $k$-th tetrahedron in the $c$-th cube. We set $v_0 = 0.01$ consistently in all of our experiments.

**Robust projection**   In Figure 12, we compare the trajectories of mesh deformaton using $\mathcal{L}_{proj}$ and $\tilde{\mathcal{L}}_{proj}$.

### A.2   MESH QUALITY

In our work, we aim to eliminate the defects of generated meshes, i.e. flipping, distortion and self-intersection. Here, we illustrate them as the preliminary. We refer to B.3 for quantitative reports on these quality statistics.

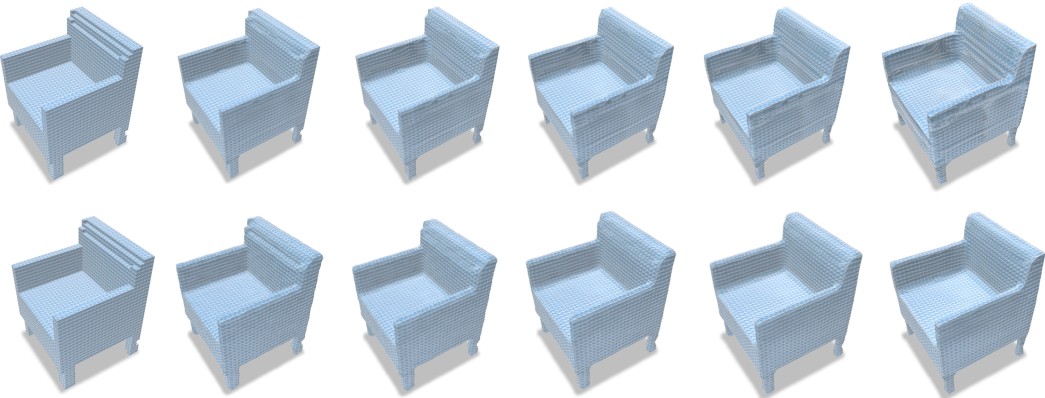

Figure 12: First row: with $\mathcal{L}_{proj}$; Second row: with $\tilde{\mathcal{L}}_{proj}$. Columns: step 0, 20, 30, 40, 60, 80.

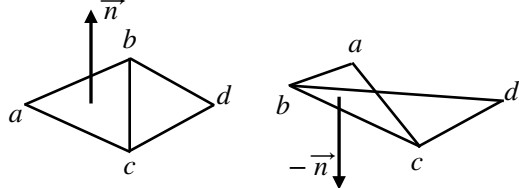

Figure 13: Left: initial regular trianglulation, Right: the triangle $abc$ is flipped. Note that the direction of the normal vector is opposite.

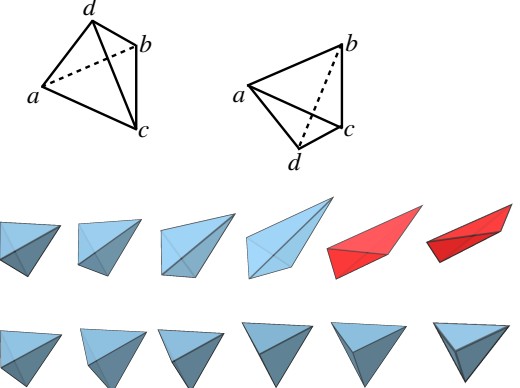

Figure 14: First row: flipped tetrahedra. Second row: a tetrahedron is deforming and flipped from blue to red. This makes the corresponding volumetric mesh illegal. Third row: A tetrahedron is deforming and preserving the orientation.

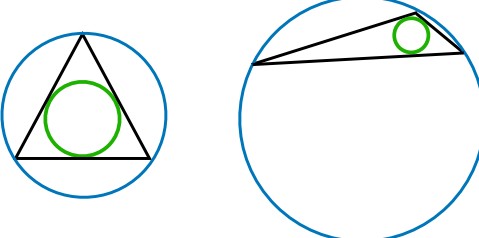

Figure 15: Green circle: inradius. Blue circle: circumradius. The left black triangle is a regular one, while the right black triangle is a distorted one.

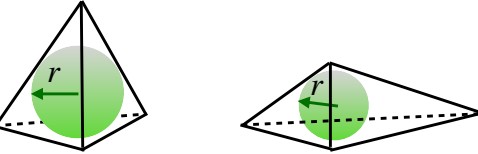

Figure 16: Green sphere: inscribed sphere of the tetrahedron with radius $r$. The left tetrahedron is a regular one, while the tetrahedron is a distorted one. A highly distorted tetrahedron will hurt the down-stream performance of the generated volumetric mesh.

**Flipped triangle**    For the surface mesh, we consider it an oriented manifold. Hence, the vertices order in each triangle face is 'wound' to keep the normal vectors pointing outwards consistently. When the vertices are deformed or mapped improperly, flipped faces occur and cause flipped normals, which will heavily harm the downstream tasks. As in Figure 13, vertex $a$ and vertex $b$ moves improperly, causing the triangle $abc$ and its corresponding normal $\vec{n}$ flipped.

**Flipped tetrahedron**    The orientation preservation for tetrahedra means that one vertex should stay on the same side of the plane determined by the other three vertices. And the sign of the determinant indicates if this same-side relation is satisfied. If the determinant is zero, the volume of the tetrahedron vanishes and the four vertices comes as co-planar. If the sign of the determinant becomes opposite, it indicates that one vertex has crossed the plane to the opposite side. As in Figure 14, vertex $d$ crosses the plane determined by $abc$, so the right tetrahedron is flipped from the left.

**Aspect ratio for triangle and tetrahedron**    For both surface mesh consisting of triangles and volumetric mesh consisting of tetrahedra, we want to make acute cells as few as possible because they tend to cause the singular matrix and large rounding errors in numerical algorithms on meshes. The aspect ratio depicts how far a cell is from the regular one. A high-quality mesh should keep most cells with a low aspect ratio. Following industrial convention, the aspect ratio for the triangle is $\frac{R_c}{2R_i}$, where $R_c$ is the circumradius for the triangle and $R_i$ is the inradius of the triangle. And we select one version of aspect ratio for tetrahedron as $\frac{h_{max}}{2\sqrt{6}r}$, where $h_{max}$ is the largest edge length in one tetrahedron and $r$ is the inradius of tetrahedron. By definition, both the perfect triangle(equilateral) and tetrahedron(all four faces are equilateral) has aspect ratio of 1. The larger the aspect ratio, the higher distortion occurs.

## B ADDITIONAL EXPERIMENTS

### B.1 ABLATION STUDY 1: NUMBER OF VOXELS

We study the setup of different voxel resolutions against the final mesh quality. We compared our current resolution with $16 \times 16 \times 16$ and $8 \times 8 \times 8$ voxel resolutions in the whole procedure. Due to the cost of the 3D convolution layer, we can not extend the configuration to higher resolutions. Thus

we show that $32 \times 32 \times 32$ is our current best. We present the result by showing the unconditional generation scores of the chair point cloud in the 1NN metric with CD and EMD. We see result in Table 3 that $32 \times 32 \times 32$ is the optimal selection comparing with lower resolution. As we say in the limitation part, our current network structure cannot fit in the GPU when scaling to a higher resolution. This may be our future direction.

| Resolution | CD | EMD |
|---|---|---|
| $8 \times 8 \times 8$ | 60.13 | 57.41 |
| $16 \times 16 \times 16$ | 57.89 | 54.26 |
| $32 \times 32 \times 32$ | **56.12** | **53.30** |

Table 3: Generation quality with different voxel resolutions. Higher resolution results in better generation quality.

## B.2 ABLATION STUDY 2: THE SAMPLING STRATEGY IN TRAINING VOXEL-CONDITIONAL NEURAL CLOSEST POINT PREDICTOR

Since we are using the deformation method rather than the Marching cube to get the surface, the main idea of our proposed trajectory sample strategy is to make the query point efficiently distributed along the displacement trajectory. Thus, we compare our sample strategy with uniform sampling, which samples the query point uniformly from the voxel shape space and Gaussian samples, which sample the points with the Gaussian distribution whose mean is the ground truth surface and the variance is 0.1 and 0.01 as IF-NET Chibane et al. (2020b) uses. We set up two choices number of query points, 75,000 (75k) and 200,000 (200k), where 75,000 is the number of query points used in our main experiment and 200,000 is the number of points used in IF-NET Chibane et al. (2020b). We evaluate the performance of unconstrained chair point cloud generation described in Section 4.1.

From Table 4, we see that our method keeps the results in both the number of query points during uniform sample and Gaussian sample performance drop in the 75,000 setups. Furthermore, from the observation, we see that sampling from the trajectory between the voxel grid to the surface closest point for our mesh deformation tasks is more efficient.

| Method | CD | EMD |
|---|---|---|
| Uniform 75k | 56.88 | 54.93 |
| Uniform 200k | 56.39 | 54.12 |
| Gaussian 75k | 56.46 | 53.97 |
| Gaussian 200k | **56.14** | **53.31** |
| Trajectory 75k | **56.12** | **53.30** |
| Trajectory 200k | **56.11** | **53.31** |

Table 4: Generation quality under different query point sampling strategy.

## B.3 ABLATION STUDY 3: THE COMPONENTS IN $\mathcal{L}(\mathcal{V})$

We perform extensive ablation studies on our method with multiple combinations of components in $\mathcal{L}(\mathcal{V})$ to illustrate their effect. In each shape class, we randomly select 500 generated voxel-represented shapes as the initialization and then apply different optimization combinations as listed in the first column of Table 5 and 6. Each final shape has its volumetric tetrahedron mesh and surface triangle mesh. For each volumetric mesh, we collect the mean/minimum/maximum tetrahedron aspect ratio(short for tetAR) and the number of flipped tetrahedra(short for tetFlip). For each surface mesh, we collect the mean/minimum/maximum triangle aspect ratio(short for triAR), the number of flipped triangles(short for triFlip) and the number of self-intersected triangles. In table 5 and table 6, the average of mean triAR and mean tetAR overall meshes are reported. And the median not average

of min/max triAR and min/max tetAR overall meshes are reported, for the sake of removing extreme numbers. Results show that the smooth term contributes a lot to enhancing every aspect of mesh quality. The orientation term removes all the flipped tetrahedra and nearly all flipped triangles. The effective noise/stochastic term helps to clean the mesh quality further. Though the one-step mapping has better quality than naive multi-step optimization, the former does not have space to improve.

| Chair | triAR (mean/min/max) | tetAR (mean/min/max) | triFlip | tetFlip | self-intersection |
|---|---|---|---|---|---|
| one-step closest-point mapping | 5.38 / 1.00 /1424.00 | 9.97/1.21/8321.54 | 69.77 | 364.83 | 61.54 |
| $\mathcal{L}_{proj}$ | 392.21/1.00/296267.87 | 111.43/1.15/140042.61 | 763.97 | 2059.52 | 3181.81 |
| $\tilde{\mathcal{L}}_{proj}$ | 35.69/1.00/19969.04 | 24.26/1.11/25485.88 | 595.93 | 1009.10 | 841.54 |
| $\mathcal{L}_{proj} + \mathcal{R}_{\mathrm{smooth}}$ | 4.45/1.00/745.69 | 35.22/1.09/20134.61 | 294.81 | 642.85 | 109.68 |
| $\mathcal{L}_{proj} + \mathcal{R}_{\mathrm{orientation}}$ | 5.01/1.00/575.28 | 4.42/1.09/55.30 | 233.57 | 0.01 | 322.33 |
| $\mathcal{L}_{proj} + \mathcal{R}_{\mathrm{smooth}} + \mathcal{R}_{\mathrm{orientation}}$ | 1.59/1.00/46.85 | 2.28/1.07/33.04 | 2.82 | 0.01 | 2.74 |
| **All** | **1.45/1.00/27.39** | **2.19/1.07/25.49** | **0.91** | **0.00** | **0.48** |

Table 5: Ablation study of how different regularization terms affect the mesh quality in chair mesh generation. All: surface + smooth + orientation + noise

| Lamp | triAR (mean/min/max) | tetAR (mean/min/max) | triFlip | tetFlip | self-intersection |
|---|---|---|---|---|---|
| one-step closest-point mapping | 3.73/1.00/677.62 | 13.10/1.17/5260.43 | 87.12 | 370.00 | 113.42 |
| $\mathcal{L}_{proj}$ | 233.24/1.00/67772.78 | 127.33/1.17/57608.62 | 505.39 | 1612.37 | 3530.89 |
| $\tilde{\mathcal{L}}_{proj}$ | 25.17/1.00/5785.62 | 44.38/1.12/15816.74 | 418.83 | 967.75 | 836.42 |
| $\mathcal{L}_{proj} + \mathcal{R}_{\mathrm{smooth}}$ | 6.77/1.00/409.83 | 29.17/1.09/10581.58 | 232.53 | 448.59 | 182.96 |
| $\mathcal{L}_{proj} + \mathcal{R}_{\mathrm{orientation}}$ | 6.81/1.00/482.14 | 5.46/1.13/66.08 | 216.07 | 0.02 | 467.91 |
| $\mathcal{L}_{proj} + \mathcal{R}_{\mathrm{smooth}} + \mathcal{R}_{\mathrm{orientation}}$ | 1.94/1.00/29.12 | 2.68/1.07/25.27 | 3.43 | 0.00 | 1.63 |
| **All** | **1.78/1.00/23.99** | **2.56/1.07/22.92** | **0.44** | **0.00** | **0.18** |

Table 6: Ablation study of how different regularization terms affect the mesh quality in lamp mesh generation. All: surface + smooth + orientation + noise

### B.4    DO OUR MODEL REALLY GENERATE NEW SHAPES?

We show that our generated model has the ability to provide new shapes rather than memorize the training set. Given a generated mesh, we retrieve the most similar mesh in the training set by computing the Chamfer distance of the point cloud samples from the generated mesh and all the meshes in the training set. In Figure 17 and Figure 18, we provide some examples of the generated chairs and lamps in a different view and compare them with the closest shapes in the training set. We observe that we are generating new shapes.

## C    ADDITIONAL DETAILS ON EXPERIMENTS

### C.1    DETAILS ABOUT PHYSICAL ROBUST MESH DEFORMATION WITH REGULARIZED PROJECTION

**Optimization step**    Empirically, we observe that 80 steps of optimization are enough for a voxel shape to converge to the volumetric mesh. Thus we choose 80 as our optimization step.

**Noise schedule of Robust Projection for distortion suppression**    In Section 3.3, we introduce Robust Projection by adding the Gaussian noise with the ratio in a linear scheduler. We want to add a stronger noise at the beginning stage and gradually decrease it at the converge stage. So, starting with an initial ratio $k = 0.1$, we decrease $k$ by 0.5 every 10 steps until 80 steps.

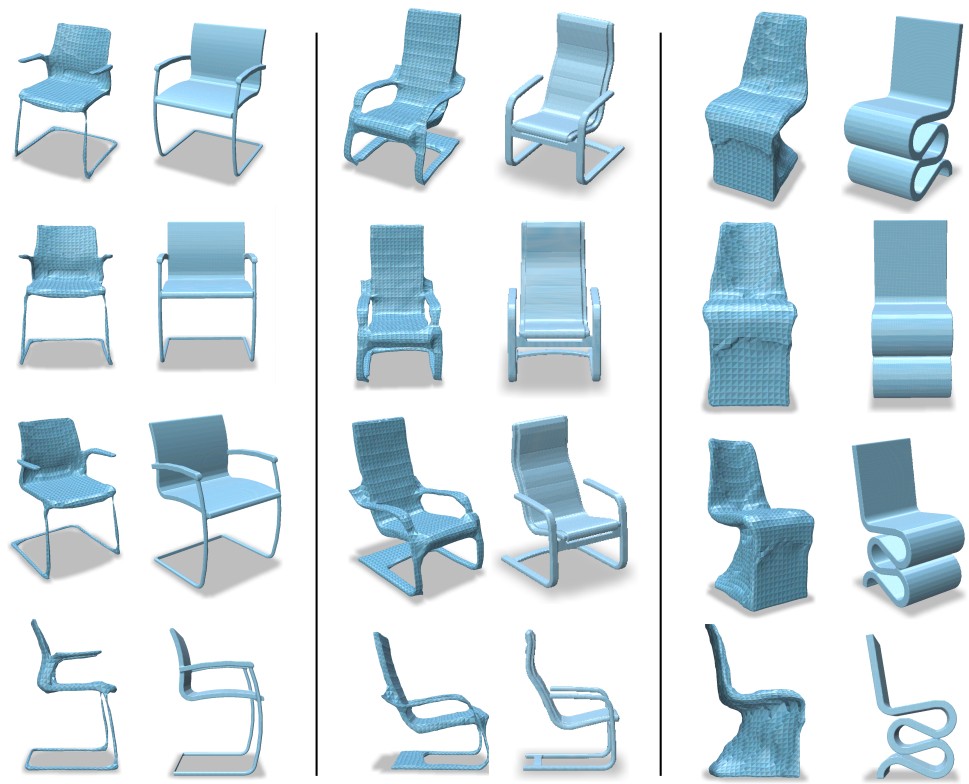

Figure 17: Generated chair samples v.s. its nearest neighbor in the training set from different views. In each column, the *left* one is the generated mesh and the *right* one is the nearest data point in the training set. We show that our model has ability to provide brand-new shape.

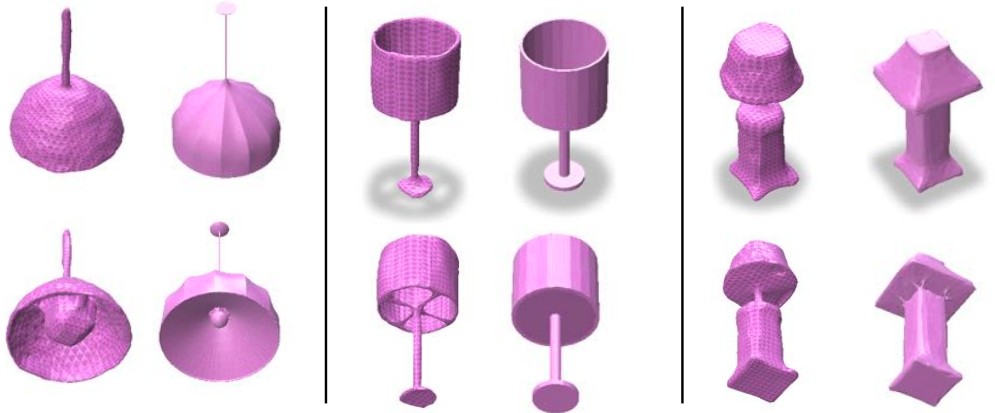

Figure 18: Generated lamp samples v.s. its nearest neighbor in the training set from different views. In each column, the *left* one is the generated mesh and the *right* one is the nearest data point in the training set. We show that our model has ability to provide brand-new shape.

**Number of query point in sample strategy**    For the query point sample strategy described in Section 3.2, we sample 75,000 query points for each training data point using the strategy we mentioned.

## C.2    NETWORK DESIGN

For DDPM model, we build up an 8-layer 3D convolution network as the denoise function. The layer width are $\{32, 64, 128, 256, 256, 128, 64, 32\}$. The first four layers gradually encode the voxel resolution from 32 to 4 by adding a stride 2 in each layer. The last four layers are 3D convolution transpose layer which recovers the voxel resolution from 4 to 32 with a stride 2 in each layer.

For Voxel-conditional Neural Closest Point Predictor, similarly, we build up a 4-layer 3D convolution network to encode the voxel with width $\{32, 64, 128, 256\}$ and gradually encode the voxel resolution from 32 to 4. As in IF-NET Chibane et al. (2020b), we apply a tri-linear interpolation on each query point based on each convolution layer's output as the multi-resolution features. After we encode the query point features based on the input voxel, we use a two-layer MLP to decode the feature into the predicted vector to the closest point.

## D    ADDITIONAL EXPERIMENT RESULT

### D.1    PERFORMANCE OF THE VOXEL GENERATOR MODEL

We also evaluate the unconditional mesh generation result in MMD and COV metrics and show the result in Table 8. We see a consistent result with the result in the main table that our generator can get a comparable or higher performance comparing with the state-of-the-art point cloud generators.

### D.2    MORE RESULTS ON ADDITIONAL CLASSES

We also evaluate lamp and bench classes and compared it with several baselines. Due to the lack of the result for some classes, we only report the result for ShapeGF and PVD and retrain these two models by ourself using the default configuration provided in the official codebases. As we shown in Table 7, we get a outstanding result compared with other methods.

| Method | Lamp | | Bench | |
|---|---|---|---|---|
| | CD | EMD | CD | EMD |
| ShapeGF (Cai et al., 2020) | 58.35 | **55.90** | 63.12 | 71.71 |
| PVD (Zhou et al., 2021) | 61.54 | 57.63 | 62.07 | 72.28 |
| NVMG | **56.89** | 57.01 | **61.99** | **70.32** |

Table 7: Generated samples performance on Lamp and Bench classes

## E    MORE SHAPE EDITING EXAMPLE

We show that our intermediate representation, the voxel is friendly for various of editing in Figure 19 and 20. We see that we can easily assemble or even directly mix two parts together to generate the smooth volumetric mesh as input.

| Method | Airplane | | | | Chair | | | |
|---|---|---|---|---|---|---|---|---|
| | MMD-CD ↓ | MMD-EMD ↓ | COV-CD ↑ | COV-EMD ↑ | MMD-CD ↓ | MMD-EMD ↓ | COV-CD ↑ | COV-EMD ↑ |
| l-GAN | 0.3398 | 0.5832 | 38.52 | 21.23 | 2.589 | 2.007 | 41.99 | 29.31 |
| PointFlow | 0.2243 | 0.3901 | 47.90 | 46.41 | **2.409** | 1.595 | 42.90 | 50.00 |
| SoftFlow | 0.2309 | 0.3745 | 46.91 | 47.90 | 2.528 | 1.682 | 41.39 | 47.43 |
| DPF-Net | 0.2642 | 0.4086 | 46.17 | 48.89 | 2.536 | 1.632 | 44.71 | 48.79 |
| ShapeGF | 0.2703 | 0.6592 | 40.74 | 40.49 | 2.889 | 1.702 | 46.67 | 48.03 |
| Vox-Diff | 1.322 | 0.5610 | 11.82 | 25.43 | 5.840 | 2.930 | 17.52 | 21.75 |
| PVD | **0.2243** | 0.3803 | **48.88** | **52.09** | 2.622 | 1.556 | 49.84 | **50.60** |
| NVMG (w/ NDC) | 0.2724 | 0.3814 | 47.84 | 50.35 | 2.491 | 1.613 | 48.23 | 48.38 |
| NVMG | 0.2314 | **0.3632** | 48.03 | 52.01 | 2.413 | **1.514** | **51.34** | 49.78 |

Table 8: Addition Experiment Result evaluted in MMD and COV.

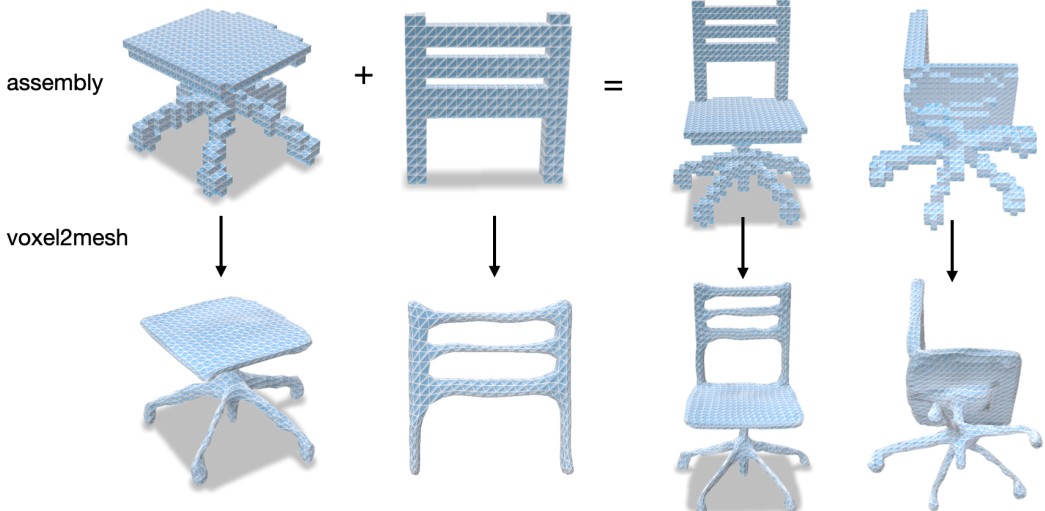

Figure 19: Parts assembly example for shape editing

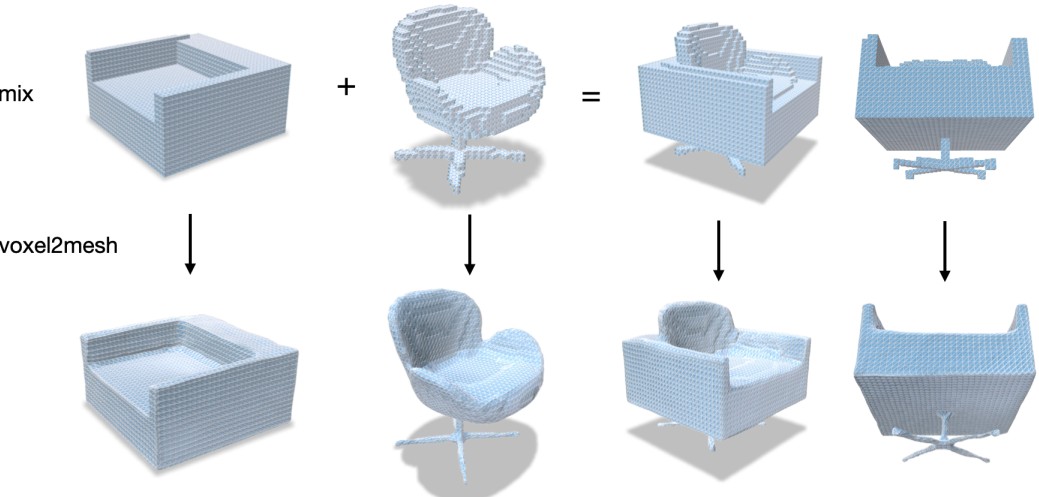

Figure 20: Parts mixture example for shape editing

## F MORE GENERATED MESHES

We show more samples of our generated results in Figure 21. We take chair, lamp, and bench as an example.

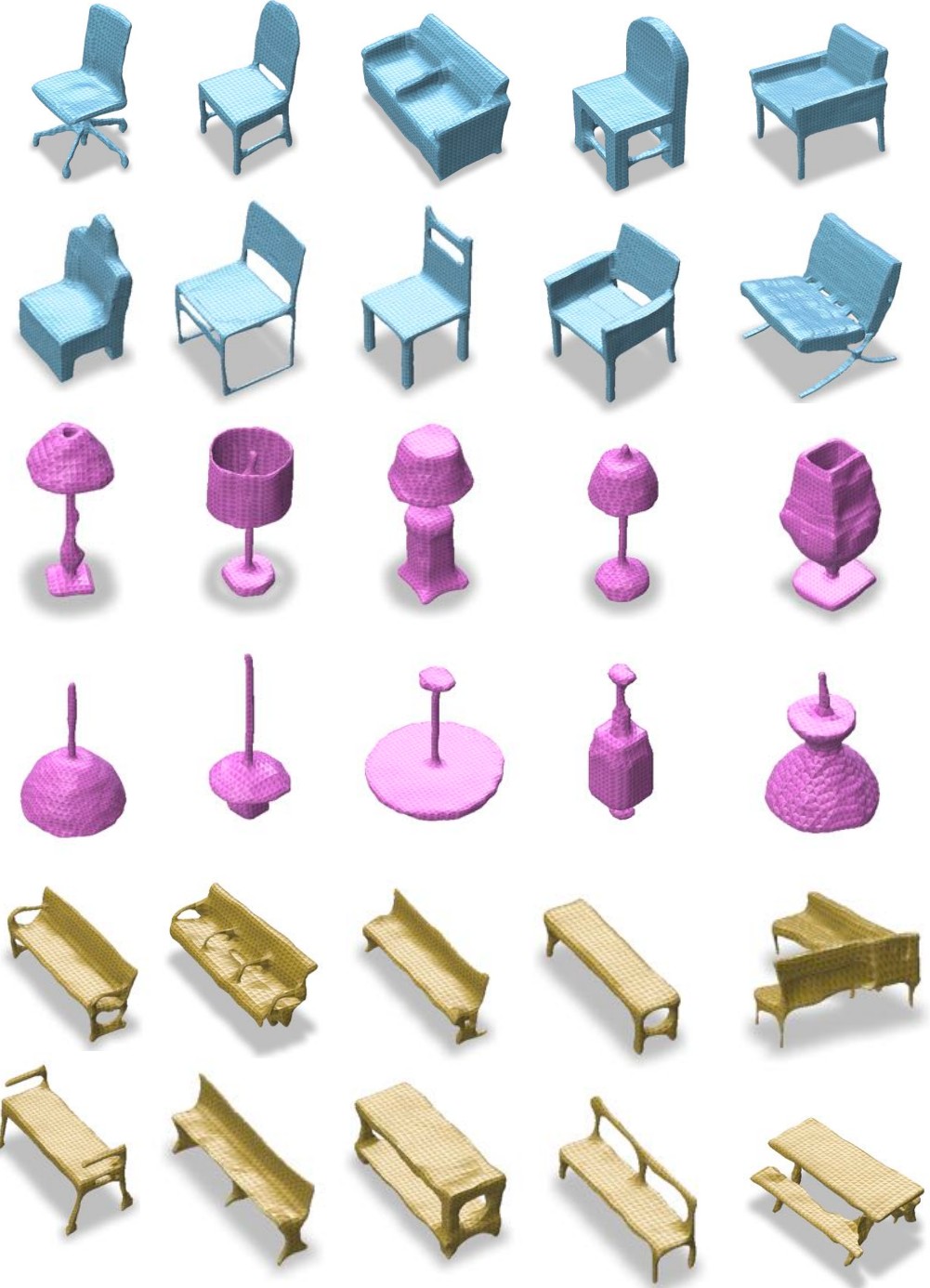

Figure 21: More generated examples

