# OpenReview forum: "Neural Volumetric Mesh Generator"
_ICLR.cc/2023/Conference — Submitted to ICLR 2023_

### Official Review · Reviewer_E4fj · 2022-10-17

**Confidence:** 4
**Correctness:** 3
**Technical Novelty And Significance:** 2
**Empirical Novelty And Significance:** 3
**Recommendation:** 6

**Clarity, Quality, Novelty And Reproducibility:**

The paper is very clear and easy to follow. The contribution is somewhat new but not that significant.

**Strength And Weaknesses:**

Strength:

- The proposed method is technicaly sound and has its novelty. Genrateing voxels using a diffusion model and then deforming the voxelized shape via optimization to get the final tetrahedral mesh is new to this field. The optimization formulated using neural network term (i.e., data term) and regularization terms looks very effective.
- The ablation study is pretty good, especially for the optimization part which contains many loss terms. As I read through the paper, I accumulated many questions in mind regarding the design choices. Most of them are well addressed in the ablation study.

Weaknesses:

- Though the method does achieve some state-of-the-art performance, technically it has some clear drawbacks including
  - The quality of the generated volumetric meshes is limited by the quality and resolution of the initial voxelized shape. In addition, the topology is determined by the voxelized shape.
  - Each module is trained/optimized separately, making the method less elegant.
- The advantage of "NVMG" over "NVMG (w/ NDC)" seems small (Table 1). The numbers might be misleading, so Figure 6 is a good example to show the actual advantage. I'd like to see more example like Figure 6 and an explanation why NDC cannot handle these cases.
- Missing timing statistics. Since the last step is an optimization process which needs to be carried out at inference, run time becomes important. How long does it normally take?
- Main results (Table 1) only have numbers for Chair and Airplane categories. I see there are additional examples for Lamp and Bench in the appendix. Why not listing their numbers in the main text? I think it's better to include them for completeness.

Additional comments:

- Sec. 3.2 first line, "a coarse representation the shape of interest" -> "a coarse representation of the shape of interest"
- Sec. 4.1 the sentence "Supp. material reports more evaluations." seems redundant.
- Sec. 4.1.1, " so in the hard cases" -> "In the hard cases"
- I'd like to see more discussion about DMtet and what's the unique advantage of the proposed method over it. DMtet also aims to generate tetrahedral meshes via learning and its representation seems to be successful in some downstream applications (https://nvlabs.github.io/nvdiffrec/).



**Summary Of The Paper:**

This paper proposes a generative model for volumetric meshes (i.e., tetrahedral meshes). The generation process has two steps: a diffusion model (DDPM) first generates a voxelized shape, which serves as a template and is then deformed to the final volumetric meshes via optimization. The loss terms for the optimization are carefully chosen to get rid of of defects like flipping and high distortion. Experiments on ShapeNet dataset shows that the method can generate high quality volumetric meshes.

**Summary Of The Review:**

Leveraging deep generative models to synthesize volumetric meshes is of course an interesting problem. Though the proposed method is far from the ideal way to do this (it's separated trained, limited by voxel template, fixed topology), I think this work is a solid step forward and the evaluation looks adaquate.

---

### Official Review · Reviewer_KpX3 · 2022-10-23

**Confidence:** 4
**Correctness:** 2
**Technical Novelty And Significance:** 2
**Empirical Novelty And Significance:** 2
**Recommendation:** 3

**Clarity, Quality, Novelty And Reproducibility:**

Overall clear, and reproducible. The main novelty consists in applying a diffusion process to a 3D grid of voxels (instead of 2D grid of pixels). But this comes at the price of a very coarse resolution, and thus poor qualitative results.

**Strength And Weaknesses:**

A strength of the paper is to employ a diffusion process to generate voxel grids. This is, to my knowledge, the first time this elegant framework is used to that end. The regularization techniques proposed for the refinement step are well ablated (smoothing, orientation, noisy reprojection)


My main concern is on the whole motivation for directly producing meshes with neural networks, as opposed to most state of the art methods (DeepSDF, DefTet, GET3D, Convolutional Occupancy Networks, …), to which authors do not compare. These methods also perform better qualitatively.
Authors only provide unclear and vague statements to support it, such as in the introduction’s first paragraph: “the implicit representation needs to be converted into explicit representations such as meshes, which by itself is not a completely solved problem”. I really don’t agree with the last part: Marching Cubes provides smooth and regular meshes off the shelf, and has been made differentiable for applications requiring to pass gradients from vertices to upstream implicit parameters (see Remelli et al. NeurIPS 202 (MeshSDF) or Atzmon et al. NeurIPS 2019).
Directly regressing meshes obviously has a very negative impact on shape quality, and requires many handcrafted regularization tricks which complexify the method. Authors need to suggest and demonstrate reasonable applications benefitting from the direct generation of meshes compared to implicit nets.

Similarly, I do not get the point of getting an inner mesh structure, whereas it is presented as a key feature of the presented pipeline - but never used for any concrete application. What is the benefit of having an inner mesh structure compared to just a surface? How to assess the quality of inner mesh structures, quantitatively? Why would a naive dense grid be bad?

If the main focus of the paper is about generative properties, then a comparison with randome latent code sampling of an implicit model (DeepSDF) is required. Authors would need to show better “sampling properties” of the proposed diffusion model.


Minor concerns are:
- the “Robust reprojection” consists in adding noise to points during reprojection. Authors show that it empirically helps, but provide no theoretical explanation for it.
- all the regularizations employed during the reprojection step are termed as “physically robust” but do not involve any physics.


**Summary Of The Paper:**

This paper proposes a generative model of meshes which are volumetric, ie. with an inner structure.
As opposed to recent implicit surfaces (based on SDF or occupancy), it directly outputs a mesh. It relies on a 2 steps process:
- 1: a diffusion model generates voxel grids.
- 2: a voxel grid refinement, using a neural network to project voxel vertices to their closest surface points.

The second step requires handcrafted regularization techniques to avoid flipped faces and preserve the mesh quality. Overall, the pipeline demonstrates good generative properties in terms of coverage.

**Summary Of The Review:**

This paper proposes to use a recent deep learning technique (diffusion) to modernize an old representation (deep explicit meshes), but it still is not on par with more modern and simpler ideas (implicit fields). Since no motivation is given for sticking to explicit meshes, I vote for rejection.

---

### Official Review · Reviewer_873W · 2022-10-24

**Confidence:** 4
**Correctness:** 4
**Technical Novelty And Significance:** 3
**Empirical Novelty And Significance:** Not applicable
**Recommendation:** 8

**Clarity, Quality, Novelty And Reproducibility:**

The description of the paper is clear, and the quality, including the writing and the results are good. The method proposed in the paper is a novel pipeline for tetrahedral mesh generation, which can be used for mesh generation, mesh editing, and mesh mixture. The method contains three modules, and a code release will be helpful for reproduction.

**Strength And Weaknesses:**

Strength:
-- This paper proposes a novel pipeline for tetrahedral mesh generation.
-- The paper has conducted adequate comparisons and evaluations, and showed generation results and editing results, which demonstrate the usability of the proposed method.
--The results are good.

Weaknesses
-- Some details need to be verified. See my comments below.
-- It seems that the voxel generation based on diffusion model adopts the existing method (Zhou et al., 2021). Is there any further improvement?
-- As for the voxel-conditional neural nearest point predictor, given the voxelized shape generated in the first stage, it predicts the nearest point on the real surface for each voxel vertex, which is equivalent to predicting the correspondence of the initial tetrahedral mesh to the real surface for further deformation. So how to deal with the ambiguity, that is, some similar models have the similar voxel representation, or even the same voxel representation. How does the network deal with this situation?
-- Although the authors give the optimization terms for mesh deformation, they do not explain how the optimization is carried out. For example, is the iteration required? what solver is used? What is the value of \lambda_a, \lambda_b and \lambda_c in equation (7)?
-- Minor issues:
In Section 3.3, ‘this naive projection does work well’ should be ‘this naive projection does not work well’
-- Missing related work:
AtlasNet: A Papier-Mâché Approach to Learning 3D Surface Generation, CVPR 2018
Learning elementary structures for 3D shape generation and matching, NIPS 2019
SDM-NET: deep generative network for structured deformable mesh, ACM TOG
TM-NET: Deep Generative Networks for Textured Meshes, ACM TOG
O-CNN: Octree-based Convolutional Neural Networks for 3D Shape Analysis, ACM TOG
Adaptive O-CNN: A Patch-based Deep Representation of 3D Shapes, ACM TOG


**Summary Of The Paper:**

This paper proposes a generator, abbreviated as NVMG, for generating volumetric meshes. With the help of the diffusion model, the voxelized shape can be generated from a randomly sampled vector, which will be divided into tetrahedral mesh. Then the initial tetrahedral mesh is deformed to the final result under the guidance of neural closest point predictor and the regularization of several carefully designed terms, including smooth term for uniform structure, orientation term to prevent defects and data term for distortion suppression. The proposed method can generate tetrahedral mesh randomly or from an input image. It can also perform shape editing through editing the voxel representation.

**Summary Of The Review:**

This paper proposes a new method with good results, although some details of the method need to be further clarified. I tend to accept this paper.

---

### Official Review · Reviewer_nDNc · 2022-10-26

**Confidence:** 4
**Correctness:** 4
**Technical Novelty And Significance:** 2
**Empirical Novelty And Significance:** 2
**Recommendation:** 5

**Clarity, Quality, Novelty And Reproducibility:**

The paper is quite clear and the work is good-quality and (to the best of my knowledge) original, modulo my comments about the magnitude of the contribution above.


**Strength And Weaknesses:**

The method is straightforward (conditioned on the DDPM which is complex but not a contribution of this paper) and appears to work quite well.

My main concern is that the contribution is quite limited. The main feature is a different way of recovering a volumetric mesh from a voxel grid. The heavy lifting in the _generation_ part is done via a prior method, viz the voxel generator. It is clear that the results are better than the ones that directly produce meshes (e.g. Fig 8), and the point that converting voxels to meshes is a better choice than synthesizing meshes directly is well taken. But the magnitude of the technical contributions beyond this point is a bit less clear. There is clear utility in having a better voxel-to-tet-mesh method. But the paper conflates this with the generation aspect and makes it harder to evaluate. Is the voxel-to-mesh method really that much better than NDC or NMC? Is this properly evaluated in a wholistic way?


**Summary Of The Paper:**

This paper proposes a method for synthesizing high-quality tetrahedral meshes. The method starts with a voxel grid generator and then produces a regualarized tet mesh from it. For the voxel grid, the authors use an existing diffusion-based model. For tet-meshing, the authors first employ a neural network trained to predict the closest point on the actual surface given the voxel grid and a query point, and then recover this surface from the closest-point function by optimizing a set of losses including regularizing losses. The authors claim that the resulting meshes are better than baselines on selected metrics, especially in terms of robustness and freedom from artifacts.


**Summary Of The Review:**

I think the paper itself is reasonably sound. My main concern is regarding the scope/magnitude of the contribution, as detailed above.

---

### Decision · Program_Chairs · 2023-01-20

**Decision:**

Reject

**Justification For Why Not Higher Score:**

While the paper has an interesting and simple idea which appears to work well, the contribution is somewhat minimal and is somewhat conflated with the overall generative modeling pipeline. The paper is missing at least one key baseline that reviewers would need to see in order to accept it.

**Justification For Why Not Lower Score:**

N/A

**Metareview: Summary, Strengths And Weaknesses:**

This paper proposes a method for synthesizing high-quality tetrahedral meshes: it uses a DDPM to generate a volumetric occupancy grid, then trains a neural field to predict the closest surface location given this voxel grid + a query point in space.

Strengths: reviewers appreciated the straightforward idea, and that the it seemed to work well (give good results).

Weaknesses: reviewers perceived a lack of novelty, as the method uses an existing DDPM approach to generate the voxel grid, so the only 'novel' component is the neural field for tet mesh extraction.

Initially, two reviewers were initially slightly in favor of acceptance, and two were opposed. After reading the other reviews and author responses, one of the in favor reviewers became more sympathetic with the opposed reviewers. The most negative reviewer was unconvinced by the author responses.


**Summary Of Ac-Reviewer Meeting:**

At the meeting, reviewers expressed concern that the paper conflates two things: DDPM voxel generation & tet mesh extraction. The first is not really a novel contribution; only the second is. The paper really only evaluates the whole pipeline as a generative model--it's hard to know how much performance is due to the DDPM vs. the tet mesh generator. The authors could reframe to focus on the tet extraction part, but reviewers expressed doubt that this would be enough of a contribution to merit an ICLR paper. If the narrative remains about a whole generative modeling pipeline, there are other important baselines that need to be considered: one that reviewers were particularly interested in was using a SotA implicit field generative model followed by an off-the-shelf (non-differentiable) tetrahedral meshing algorithm e.g. TetGen or TetWild.

Given these concerns, I am recommending the paper be rejected.

Finally, a suggestion for resubmission: having the title/narrative be about "mesh generation" primes reviewers to think that the paper is going to be about a neural net that manipulates meshes, when it really isn't. Authors might consider renaming the paper / reworking the narrative so that reader expectations are better aligned with what the method actually does.